# Environmental Impacts of Thermal and Brine Dispersion Using Hydrodynamic Modelling for Yanbu Desalination Plant, on the Eastern Coast of the Red Sea

**Naif S. Aljohani** [1,2], **Yasar N. Kavil** [3], **Puthuveetil Razak Shanas** [4], **Radwan K. Al-Farawati** [3], **Ibrahim I. Shabbaj** [1], **Nasser H. Aljohani** [2], **Adnan J. Turki** [3] **and Mohamed Abdel Salam** [5,*]

1. Department of Environmental Sciences, Faculty of Meteorology, Environment and Arid Land Agriculture, King Abdulaziz University, P.O. Box 80208, Jeddah 21589, Saudi Arabia; nssh56@gmail.com (N.S.A.); ishabbaj@kau.edu.sa (I.I.S.)
2. Saline Water Conversion Corporation, P.O. Box 5968, Riyadh 11432, Saudi Arabia; naljohani2@swcc.gov.sa
3. Department of Marine Chemistry, Faculty of Marine Sciences, King Abdulaziz University, P.O. Box 80207, Jeddah 21589, Saudi Arabia; fidayas@gmail.com (Y.N.K.); rfarawati@kau.edu.sa (R.K.A.-F.); aturki@kau.edu.sa (A.J.T.)
4. Physical Oceanography Division, National Institute of Oceanography, Dona Paula, P.O. Box 403004, Panaji 403001, Goa, India; shanaspr@gmail.com
5. Department of Chemistry, Faculty of Science, King Abdulaziz University, P.O. Box 80200, Jeddah 21589, Saudi Arabia
* Correspondence: masalam16@hotmail.com; Tel.: +966-541886660; Fax: +966-2-6952292

**Abstract:** For any coastal desalination plant, the most effective and practical way to dispose of their brine is to thermally discharge it into the sea via outfalls at some distance from the coast. This study focused on the environmental impacts associated with brine and thermal discharge arising from seawater desalination plants at Yanbu, Saudi Arabia, on the southeastern coast of the Red Sea. The impacts associated with recirculation patterns and dispersions were investigated with the calibrated three-dimensional numerical model Delft3d. The environmental impact assessment and the process of identification and characterisation could help improve strategies for better planning and management of the technological solutions related to desalination. Analysis of the model simulations for the different seasons also suggested that around the outfall location, the magnitude of the flow was always high when considered together with the presence of seasonal eddy circulations. Although the tidal flow is lower, the ambient current and wind cause the far-field discharge to spread along the north–south direction during the winter and summer. The thermal and brine dispersion and environmental compliance were assessed in terms of the extent of dispersion. The well-mixed environment caused more rapid dispersion. From the impact level assessment perspective, the study indicated rapid dilution and dispersion of the wastewater at the study region. The present offshore outfall and further offshore locations were far enough to ensure quick dispersion.

**Keywords:** desalination; hydrodynamic modelling; dispersion; Red Sea

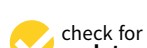



## 1. Introduction

Recent advances in desalination technologies have significantly minimised the environmental impacts associated with desalination and reduced the consumption cost. The degree of waste source and effluent discharge processes have generally shown an extensive role, particularly for coastal settings, in impact on the environment and the general need to comply with local environmental criteria and regulations. However, brine disposal has adverse impacts on the coastal and marine environments, as well as on the process of modification of salt and heat content of the ocean, and it could be a potential threat to the system, as it carries harmful substances [1]. The essential parts of the environmental assessment objective herein were to reduce the negative impacts of treated water disposal and

to substantially reduce the cost to optimise the discharge systems without discharge control [2]. Information on the hydrodynamic condition of the region is very important in the design and construction of any coastal structures and developmental activities. A proper investigation and understanding of the existing hydrodynamic setup helps in designing structures according to the complexity of nearshore processes in the region. Numerical simulations and predictions have been widely used by researchers and engineers as an effective approach to optimizing design according to the environmental condition of the region.

Generally, the brine discharge effluent from desalination plants could have salinity twice that of normal seawater, and it poses a great threat to the environment into which it is released. [3–6]. Most desalination plants lie along the coastline, and the effluent generally discharges or releases into the sea with the assumption that this ensures quick dispersion. Thermal effluent from desalination plants generally has elevated temperatures compared with the adjacent region. This does not alter the mean condition of the environment. However, the introduction of thermal effluent can significantly alter the flora and fauna in the water via the steep variation of temperatures at the releasing area [7]; this in turn impacts the dissolved oxygen considerably, which suppresses the metabolism and reproduction and negatively affects the growth of the aquatic ecosystem. Especially for the thermal dispersion from power plants, there have been many studies conducted globally and in adjacent marginal sea regions such as the Arabian Gulf and Mediterranean Sea [8–12]. Prediction of the thermal plume has been carried out through the development of a numerical model in [11,13]. Ref. [14], a study at the Bohai Sea, underscored the importance of thermal dispersion and meteorological effect through a 2D numerical model. Ref. [12] implemented three-dimensional modelling for the simulation of water temperature distribution. There have been several studies that reported the use of commercial tools as the key tools for the simulation of thermal dispersion in nearshore coastal environments, estuaries, and lakes [15–18].

Refs. [15,16] analysed thermal dispersion simulation results with the use of the MIKE software. In the Red Sea and Lake Ontario, a finite element model was used to simulate the thermal dispersion characteristics [17,18]. Thermal dispersion at the Bohai Sea was simulated by the COHERENS model [19]. Refs. [20–22] implemented a 3D hydrodynamic modelling tool to study the temperature dispersion characteristics over the Gulf of Suez region, the Gulf of Mexico, and Delaware Estuary. Changes in the wind can cause large changes in the mixing; consequently, the water temperature could mix in a vertical direction. Recently, using the Delft3d model (Deltares, Netherland) at the study area, some preliminary analysis of the environmental impact assessments of circulation characteristics at the desalination plant near the Yanbu region was reported in [23]. Globally, there have been many studies carried out based on numerical modelling to study the dispersion characteristics of thermal and brine discharge from desalination plants [24–27]. Specifically, for the Red Sea coast, the dispersion of far-field characteristics, the extent of dilution, and the hydrodynamic circulation pattern near the disposal location were characterised with the help of a 3D numerical model [23]. Although [23] reported the general characteristics for the study area, a detailed investigation of thermal and brine discharge needs to be conducted. The recirculation of temperature and salinity are considered the important design parameters in the structure of the intake outfall position and in the desalination plant unit. Numerical models are usually used to study such recirculation and to choose the most suitable intake outfall design, minimise the recirculation, and study the advection and dispersion of brine discharge from outfalls. Furthermore, numerical models are important tools used to study the impact of marine outfall on marine ecology. This requires the modelling of salinity and some water quality parameters in the surrounding area of the disposal outfall location.

Coastal areas cover about 70% of the cities in the world; the dynamics of the coastal process and the complexity of the nearshore environment must be well understood. In the current study, a hydrodynamic simulation was carried out for the Yanbu region, Saudi

Arabia, on the eastern coast of the Red Sea, using the Delft3D modelling software package and based on continuity, momentum equations, and temperature and salinity constituent relations.

The current study aimed to:

- set up a hydrodynamic simulation for the Yanbu Desalination Plant region by using boundary values from a regional domain simulation;
- carry out the hydrodynamic simulation by taking both tidal and wind-induced circulation into account;
- study the circulation pattern for the region in different seasons (winter and summer);
- identify the impact of thermal and brine waste discharge dispersion through numerical modelling.

## 2. Study Area

Yanbu is one of the main industrial areas of Saudi Arabia, situated in the Al Madinah Province of the Kingdom of Saudi Arabia on the shores of the Red Sea (Figure 1). The Yanbu desalination plant is situated on the Red Sea Coast. Due to the area's industrial applications, the on and off region of the port area will be affected by heavy metal discharges and inorganic and organic waste disposals. The Yanbu desalination plant is 42 km south of Yanbu city, 145 km west of Medina, and 40 km west of the Badr Governorate. The plant's estimated capacity is 438,000 m$^3$ of water per day, in addition to producing 3024 MW of electricity. The Yanbu desalination plant is regarded as one of the most important projects in the area, supplying the Kingdom with large quantities of high-quality water while also producing electricity as a by-product and thus acting as a low-cost source. The plant started operation with MSF technology, and then was expanded to increase its capacity to reach more than 438,000 m$^3$/day using three technologies (MSF, MED, and RO). The operation line of potable water consists of an intake system, a chlorination system, a screen, a backwashing system, chemical treatment, a boiler system, desalination units, a potabilisation plant, and turbines. Currently, the plant produces 212,184 m$^3$/day.

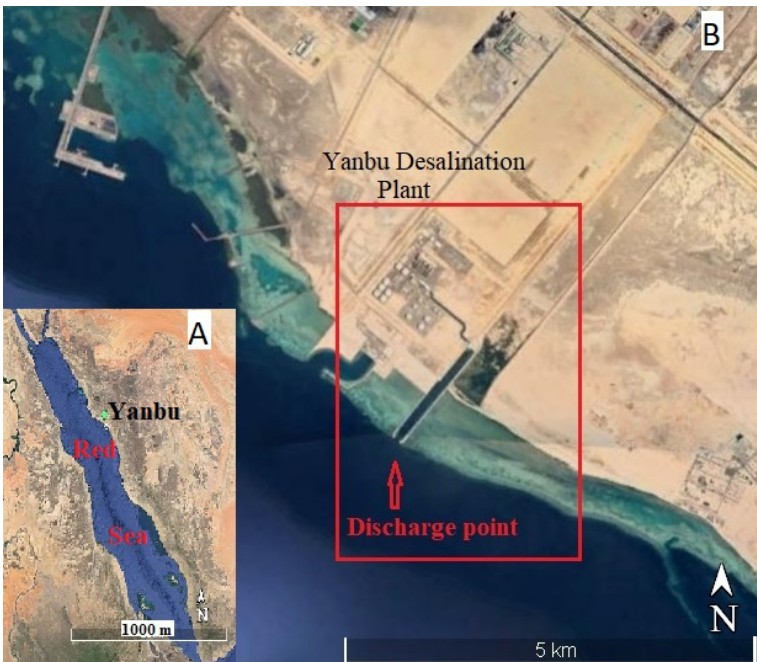

**Figure 1.** Study area; (**A**) location of Yanbu along the Red Sea, (**B**) location of the Yanbu desalination plant along the Yanbu coast.

## 3. Methodology

### 3.1. Numerical Model General Description

Recent advances in modelling have increased dramatically such that it can be effectively used as a tool for many environmental issues [28]. These environmental issues include predicting the flow pattern of a region and the dispersion of pollutants in water systems, quantifying the amount of sediment that has been transported at a region of interest, etc.

The basic structure of a model yields an understanding of the hydrodynamic characteristics of the region it is applied to. The flow module simulation will serve the needs of this study. The separate modules for waves, hydrodynamics, and morphology makes the model identify the different processes extensively. The brine discharge, sediment, and flow characteristics generally form the basic needs of the assessment to be carried out.

Delft3D is a 3D dimensional finite difference numerical modelling system [29] and is widely used for studies related to thermal and brine discharge [30,31]. Delft3D is a flexible, integrated modelling framework developed by Deltares, which simulates two- and three-dimensional flows, waves, sediment transport and morphology, water quality, and ecology and is capable of handling many of the interactions between those processes. The hydrodynamic model is capable of running in both Cartesian and spherical coordinate systems, and it is implemented with regard to the tidal force, intertidal drying and flooding, the distribution of temperature and salinity due to the density gradient, eddy circulation, turbulence closure, etc.

Different modules are available in the Delft3d system, where the Delft3d-FLOW module constitutes the hydrodynamic module and serve as the basis for the water quality and particle tracking, waves, and morphology modules. A far-field dispersion study is often coupled with the water quality module named Delft3d-WAQ. Further, it is coupled with the Lagrangian particle tracking module for the simulation of the non-steady modelling of midfield water quality analysis.

### 3.2. Setting up the Model for the Area

The investigation is carried out with different modules of Delft3d. The modelling system is capable of computing the hydrodynamic characteristics, wind waves, and transformation sediment dispersion and transport. Figures 2 and 3 show the domain, bathymetry, and rectangular curvilinear mesh used for the model simulations. The model simulations are based strictly on the inputs that describes the bottom depth and coastal shoreline condition. The forcing conditions of the prescribed boundary and initial water level, tidal characteristics, waves, etc., have a direct influence on the simulation. The hydrodynamic module from the Delft3D software package will be used for the hydrodynamic simulation of the Yanbu region. The modelling will be carried out for the regional domain (Red Sea) and the local domain (Yanbu). The offshore boundary conditions for the local domain will be extracted from the regional numerical simulation.

The model spin-up time is considered until the outputs look stable. The boundary at the offshore seaward direction is set to zero. Similarly, the model started with a water level of 0 imposed as the initial boundary condition.

### 3.3. Numerical Modelling

The hydrodynamic module from Delft3D solves the advection–dispersion–reaction equations. It is implemented with an implicit scheme fitted with the k-$\varepsilon$ turbulence closure method. The model setup can be either Cartesian or spherical. The model works by taking account of the tide generating force, the constituent distribution, the viscosity concept, spatially varying meteorological inputs, etc.

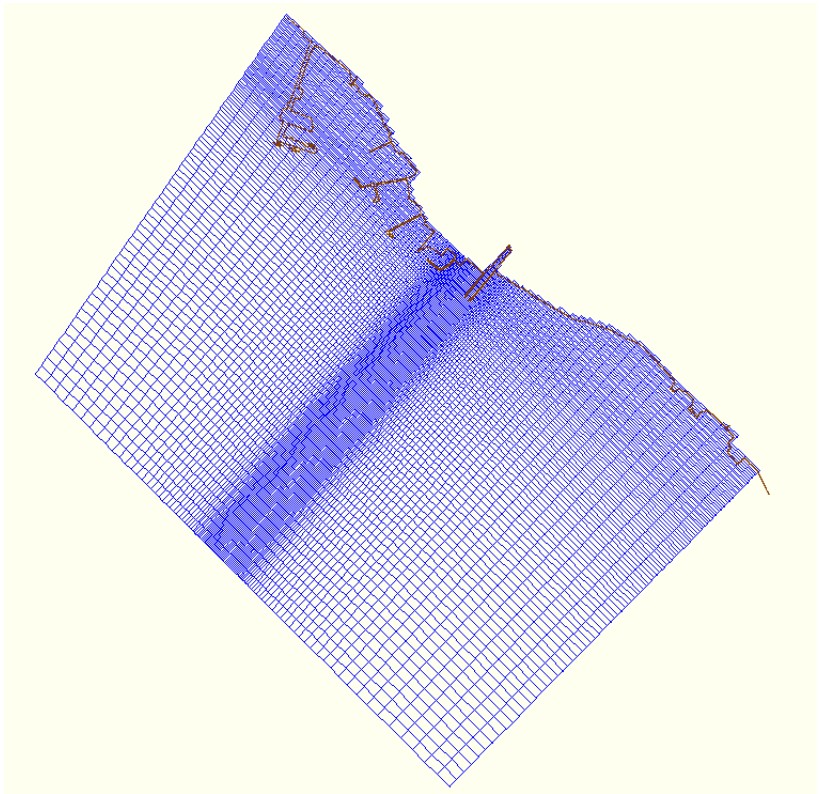

**Figure 2.** The model domain and grid used for the model simulations.

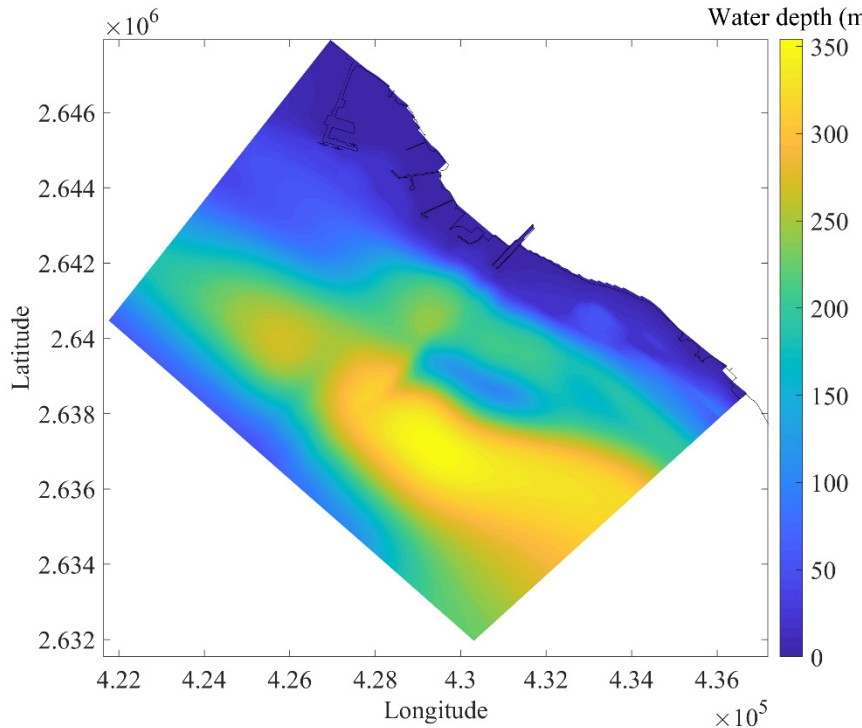

**Figure 3.** The model domain and bathymetry used for the model simulations.

### 3.4. Model Set Up

A 3D hydrodynamic model has been set up to cover an area of about 6 km along the shoreline of the proposed area and extending 5 km offshore, to sidestep any effects of the boundary on the model results. Tide data at the boundary will be applied to ensure that an accurate reading of the water circulation is included in the computational domain. The atmospheric data were obtained from the reanalysis product of CFSR using a spatial resolution of 0.2 degrees at every hour interval [32].

Since both tidal and meteorological influence operates on the hydrodynamic simulation, it is necessary to incorporate both inputs for the model to run. Hence, the model simulations were carried out by forcing the open boundaries with tidal elevations and by forcing the surface boundaries with the CFSR wind velocities [33]. The tidal boundaries, as elevations at the open boundaries of the local model (Yanbu), were derived from the regional model. The circulation pattern for different seasons is investigated. Besides this, the salinity and temperature dispersion were also explored.

### 3.5. Boundary Conditions

A Riemann boundary condition is used at the open boundaries. The major tidal components were extracted from the global ocean tidal model TPXO 7.2 on a $1/4 \times 1/4$-degree resolution global grid and were prescribed at the boundary cells and linearly interpolated. To reduce the bottom friction, 100 m$^2$/s is used as the Chezy coefficient. The density is set at $\rho w = 1023$ kg m$^{-3}$, and observation points are randomly selected in both the cross and alongshore directions at the disposal location. We have considered several observation points, and used them for the estimation of the extent of the dilution and other properties of the discharge water.

To simulate the dispersion characteristics, it is necessary to predict the hydrodynamical characteristics of the region as driven by the meteorological forcing along with the tidal input. The Delft3d hydrodynamic module served as the source of the flow characteristics. The coordinate system used in the model is the σ coordinate approach, which depicts that each layer's thickness is a percentage of the local water depth. Five layers were implemented for the area as it is of shallow depth.

## 4. Winds

10 m wind data from [32], available at an hourly temporal resolution with a spatial resolution of $0.2° \times 0.2°$, has been used in this study. The climatological characteristics can be seen from the wind rose plot. (Figures 3 and 4). The winds are predominantly from the NNW direction at the offshore location. At the nearshore location, the wind rotates in almost all directions in the upper half from west to east; however, the higher wind speeds are from the NNW direction.

### Climatology of Surface Wind

To understand the hydrodynamic nature of the location, it is necessary to have an understanding of the location's pre-existing meteorological conditions. The climatology of the wind will provide a clear picture of the station under consideration. Usually, this analysis has been carried out with wind rose. The analysis has been made using Climate Forecast System Reanalysis (CFSR) 38-year-old reanalysis wind. The same applies to the tidal constituents describing the basic tidal cycle derived from measurements in a station. The wind rose indicates the frequency of occurrence of winds from a different direction for the number of wind speed ranges. The wind pattern at the study area has been analysed, and it is found that most of the winds (80%) are from the west and northwest directions, although winds from other directions are present (Figure 4). The W winds dominate the study area at the offshore point, whereas the wind direction shifted west at the nearshore location (Figure 5).

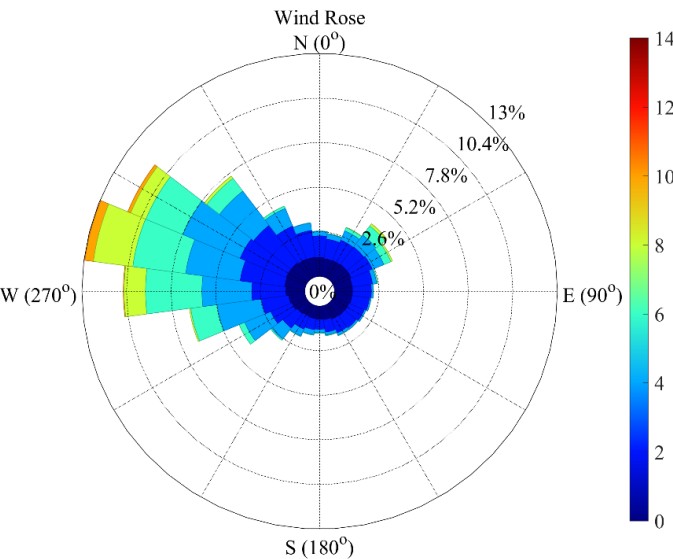

**Figure 4.** The wind rose diagram at the nearshore location.

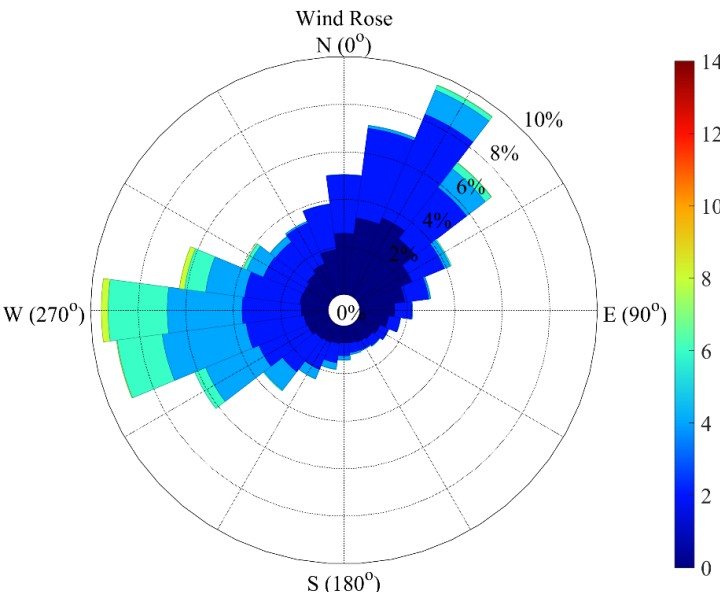

**Figure 5.** The wind rose diagram at the offshore location.

The daily land/sea breeze system is greatly influenced by the normal flow condition near the coast. This wind phenomenon is controlled by large temperature gradients between land and sea and causes strong east/west winds all along the Saudi coast. The influence of land/sea breezes is believed to decrease gradually further offshore until a distance of approximately 50–100 km from the coast. Therefore, the influence on the flow and water level conditions is restricted to the nearshore region. The climatology of the wind also suggests that westerly winds are dominant in the region, with an average wind speed of around 4 m/s. The higher winds observed during winter and summer rarely go above 15 m/s. Hence, the currents will be inconsistent with the winds, and as these storm winds approach the ambient seawater, temperatures are high, and thermal dispersion to sensitive shallow areas or other intake facilities should be prevented or limited.

## 5. Circulation Features

The hydrodynamic circulations in the study region have been simulated covering most of a spring–neap tidal cycle for two seasons, namely, summer and winter. Here, the

month of February represents the winter, and the simulations for the month of August correspond to the summer simulations. The water depths in the study region are relatively shallow, hence five vertical layers are taken into account in the model simulations. The depth-averaged current speeds and tidal ranges are discussed here.

Figure 6 shows the current patterns (flood and ebb currents) during the spring tide. The ebb currents are situated towards the north and northwest, where stronger currents are observed in the vicinity of the discharge location. Relatively weak currents are noted in the offshore regions. The flood currents are situated towards the southeast and are stronger than the ebb currents due to the effect of winds going in the same direction. In the vicinity of the plant outfalls, the ebb currents are significantly high.

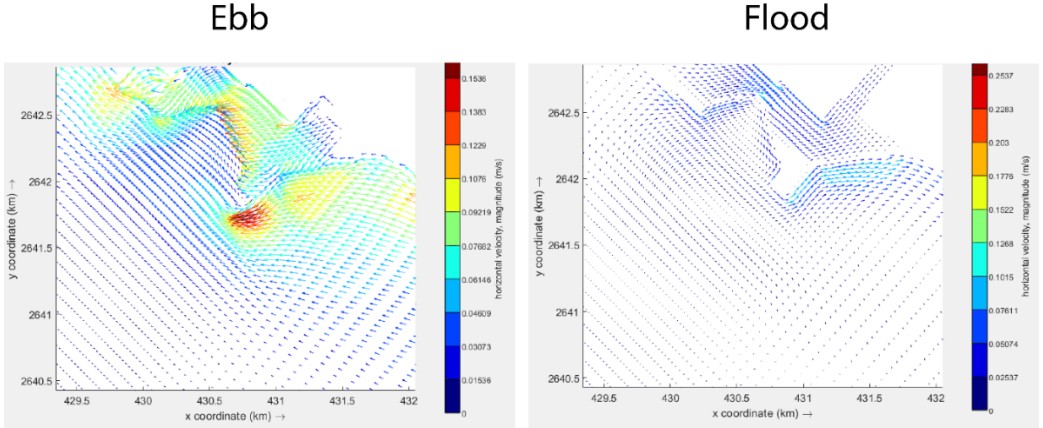

**Figure 6.** The typical nearshore ebb and flood condition over the study area.

*Depth Average Current Speed and Direction during Summer and Winter*

The hydrodynamic simulation has been carried out using the Delft3d-FLOW module. The setup of the model is described in detail in the model description section.

Figure 7 indicates the depth-averaged flow characteristics during the summer period. The general flow pattern consists of a southward flow in the offshore region, and the surface water near the Yanbu area tends to flow towards the south. A current speed on the order of 20 cm/s is often observed at the offshore location, whereas the current speed is generally stronger in magnitude near the coastal region (Figure 8). The depth-averaged current speed for winter is shown in Figure 9 below. The depth-averaged flow shows a relatively higher magnitude compared to that for summer. Furthermore, the strong nearshore currents favour a wastewater discharge at a higher dilution rate (Figure 9).

Figure 10 shows the current speed with the presence of eddies near the discharge point. The presence of anticyclonic local eddies such as gyre near the discharge point during the winter season is noticeable, enhancing the dilution at a faster rate than in summer.

The seasonal depth-averaged current pattern is shown below. During winter, the currents are situated towards the SSE and NNW directions, with a relatively high magnitude for the SSE currents. The current speeds are relatively stronger in the nearshore location, while they are lower at the offshore waters of the study location. Along the northern part of the study area, the condition is slightly weaker where the currents are situated towards the SSE direction. The weakest offshore currents are observed during the summer period and the strongest during the winter, whereas stronger nearshore currents are observed during both summer and winter. Clear evidence of eddy circulation is observed during winter, especially at the Yanbu discharge location. The divergence of the currents at the nearshore location is also shown in Figure 10. In such cases, the dispersion is spread both north and southward from the discharge location.

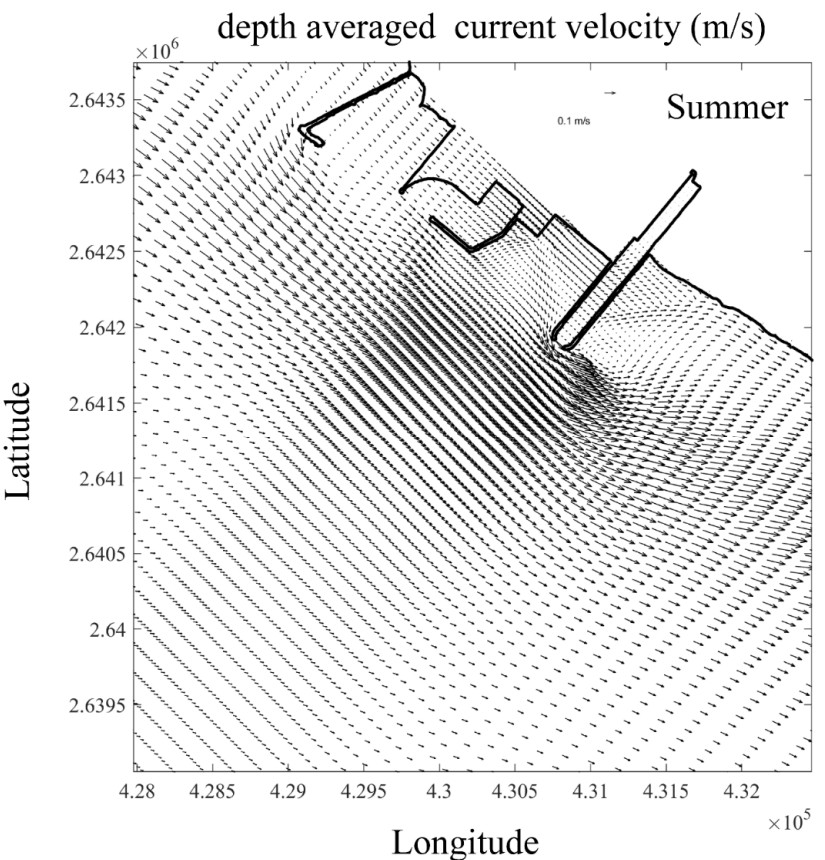

**Figure 7.** Depth-averaged current speed at the nearshore area during summer.

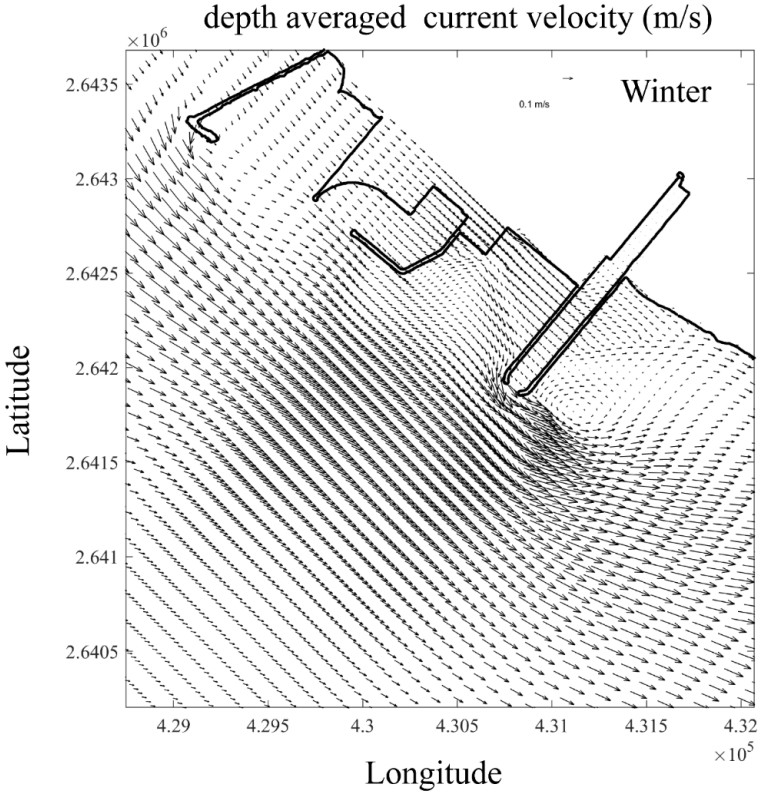

**Figure 8.** Depth-averaged current speed at the nearshore area during winter.

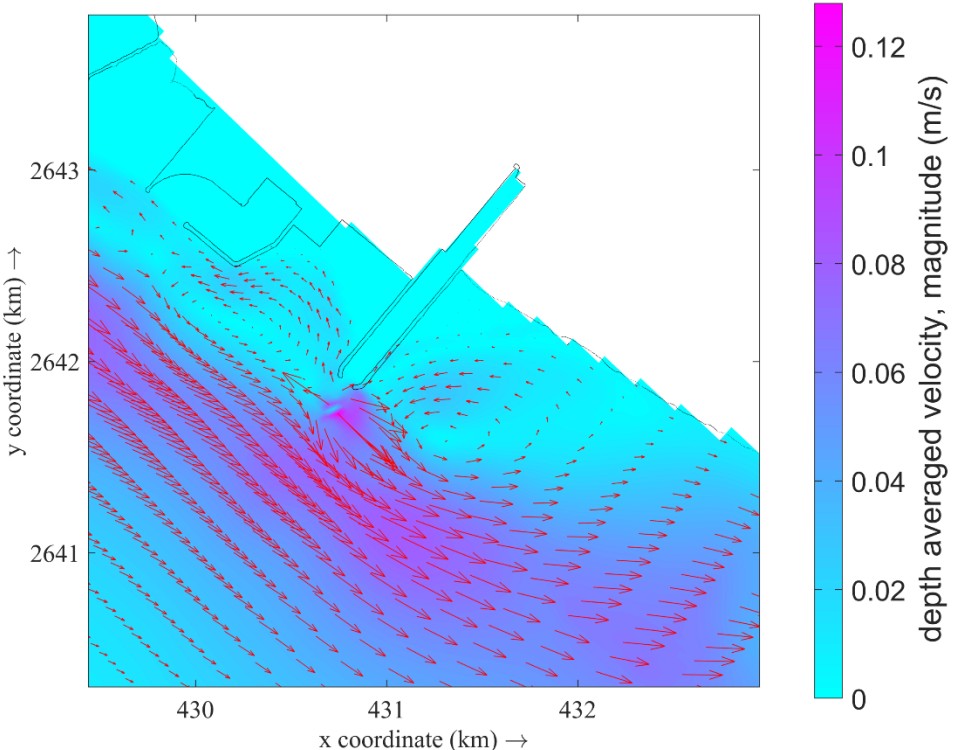

**Figure 9.** The typical current speed condition with the presence of eddies near the discharge point.

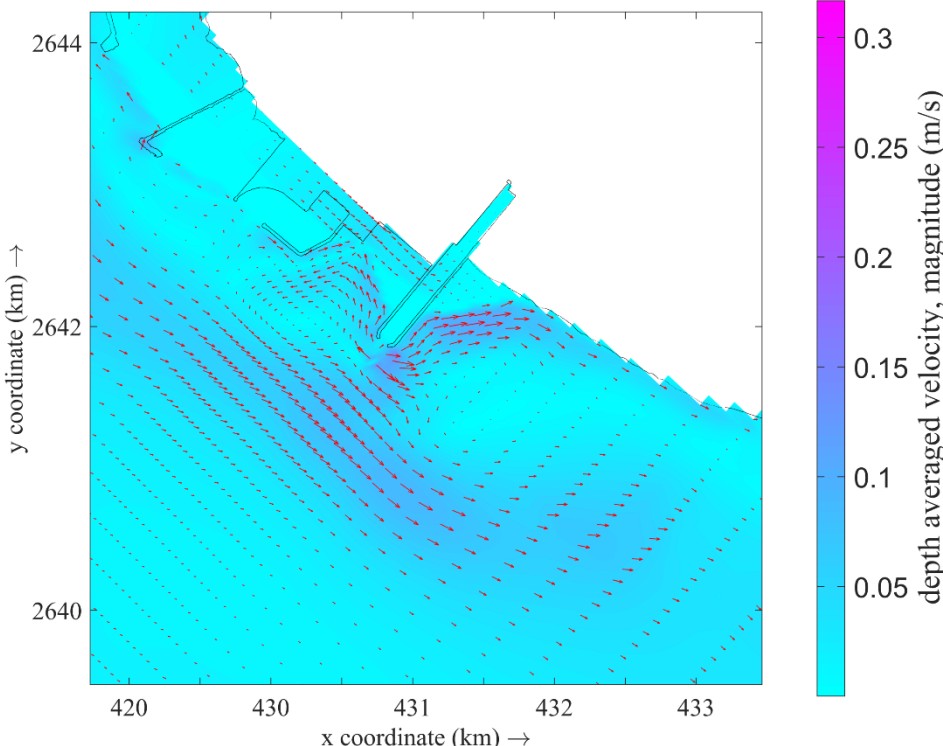

**Figure 10.** The typical divergence pattern of the current at the nearshore region.

## 6. Tides

The simulated water level at the finer grid observation point is shown below for summer and winter. Water level analysis is based on the harmonic analysis method of least square. The prediction of the tide is completed with harmonic analysis [33]. The product of the tidal analysis is shown in Figures 11 and 12, as well as Table 1.

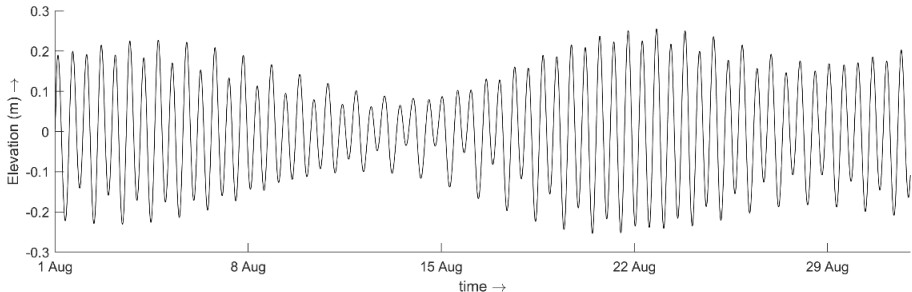

**Figure 11.** The nearshore water level at the representative point in the finer grid model during summer.

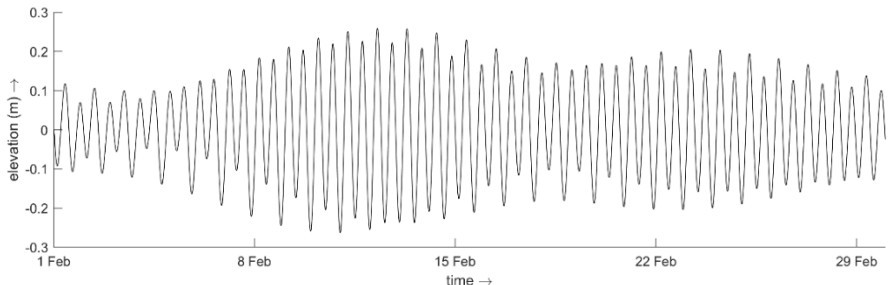

**Figure 12.** The nearshore water level at the representative point in the finer grid model during winter.

**Table 1.** The amplitude and phase of the major tidal constituents at the observation point.

| Constituents | Amplitude | Phase |
|:---:|:---:|:---:|
| M2 | 0.1645 | 121.29 |
| S2 | 0.040 | 157.3 |
| N2 | 0.0533 | 92.26 |
| K1 | 0.0253 | 141.07 |
| O1 | 0.0100 | 98.5 |

Figure 13 shows the time series of the current speeds at the discharge location and further offshore. The current speeds are relatively stronger in the nearshore location, while they are relatively lower in the offshore region. The diurnal variability in the current speeds is due to the influence of winds, as presented in Figure 14. During winter, the current speeds are relatively higher near the discharge location and are found to be consistent during both seasons.

Velocity at the discharge point (Summer)

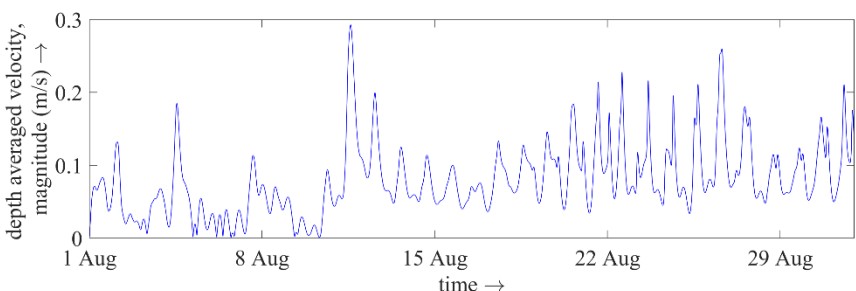

Velocity at the discharge point (Winter)

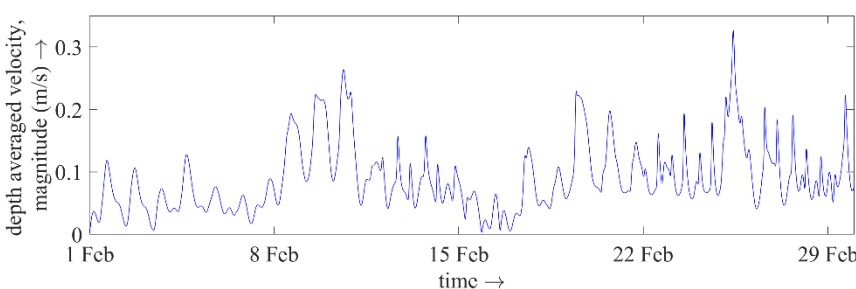

**Figure 13.** The velocity at the discharge point during summer and winter from the model outputs.

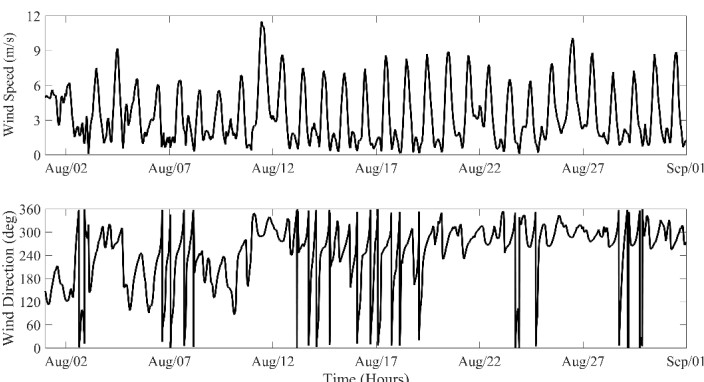

**Figure 14.** The wind field at the discharge point.

## 7. Thermal and Brine Dispersion

The temperature dispersion at the Yanbu coast has been analysed considering strong and calm wind conditions. The thermal and brine discharge at the nearfield areas are presented. The high saline and hot water discharge during phase 3 at the outfall will affect the stratification, mixing, and advection–dispersion processes in the vicinity of the discharge location. In addition, the ocean currents caused by horizontal and vertical gradients in the seawater density will play a major role in the recirculation of the seawater between the outfall and the intake. The so-called discharge from the desalination plant is hypersaline, without any chemical known as brine used in the process.

Depending on the regional hydrodynamic settings and the prevailing conditions, the brine discharge can have a significant impact along the coast under consideration. High priority should be given to the salinity and thermal discharge. Nevertheless, a salinity and temperature higher than the ambient conditions needs to be investigated using a 3D hydrodynamic model.

Different scenarios were developed for the region based on the flow characteristics (Table 2). Such investigations are required to detail the differences in meteorological impact and the impact of the flow on dispersion.

**Table 2.** Modelling scenarios for the potential range of environmental outcomes.

| Seasons | Number Scenarios | Flow rate (m³/h) | Wind Speed (m/s) | Salinity | Temperature |
|---|---|---|---|---|---|
| Summer | Scenario 1 | 250 | Calm (4 m/s) | 44 PSU | Ambient |
| | Scenario 2 | | Strong (10 m/s) | | |
| | Scenario 3 | 250 | Calm | Ambient | 10 degree excess |
| | Scenario 4 | | Strong | | |
| Winter | Scenario 5 | 250 | Calm | 44 PSU | Ambient |
| | Scenario 6 | | Strong | | |
| | Scenario 7 | 250 | Calm | Ambient | 10 degree excess |
| | Scenario 8 | | | | |

The spatial plots of the average increase in surface temperature and salinity in the near- and far-field areas are presented in Figures 15–18. These excess surface temperature and salinity plume plots are used to evaluate the thermal dispersion of the discharge water and its impact on near- and far-field areas.

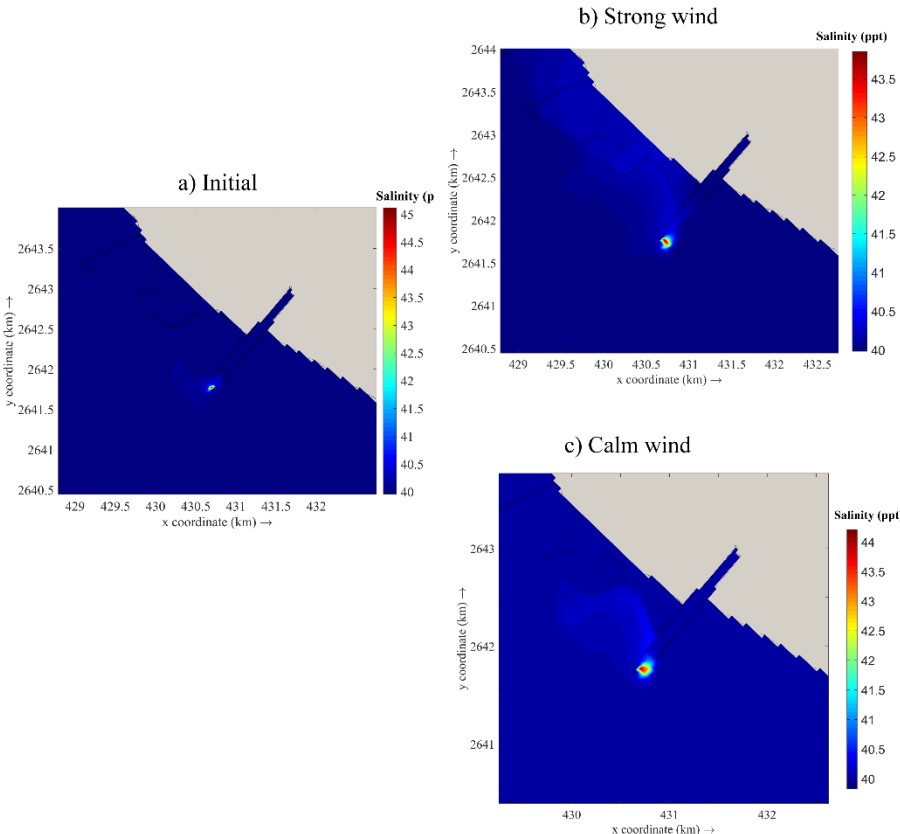

**Figure 15.** The brine plume discharge (**a**) initially and (**b**) after dispersion during strong wind conditions and (**c**) during calm wind conditions.

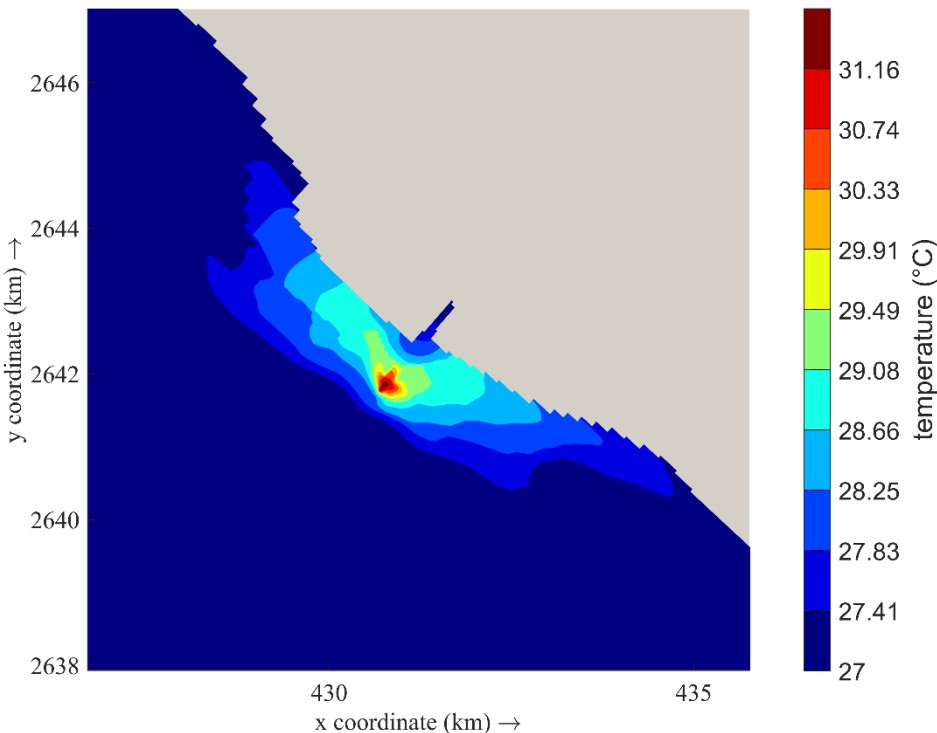

**Figure 16.** The thermal dispersion for a strong wind condition.

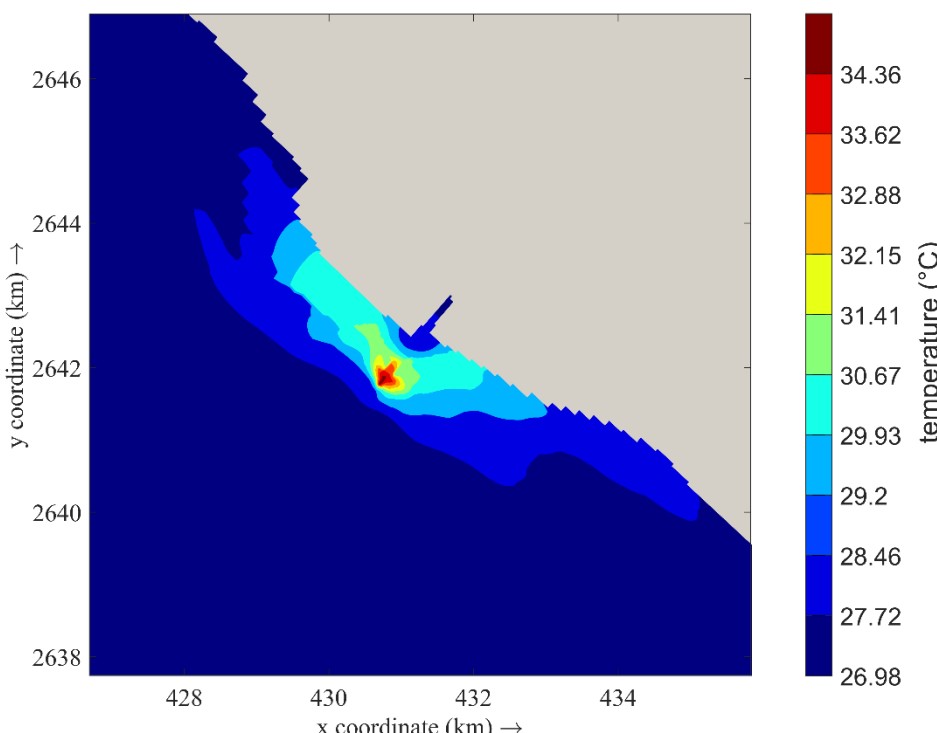

**Figure 17.** The thermal dispersion pattern for a calm condition.

The ambient salinity in the offshore waters is around 40 PSU. Figure 15 show the mean salinity distribution in the study region for the strong wind condition and the calm wind condition. In view of the given outfall salinities, the relatively high salinity plumes are visible in the outfalls, which ranges up to 44 PSU. However, the given discharge salinity in the outfall location is reasonably low (44 PSU). The salinity dispersion suggests that the salinity never goes above 2 PSU at the discharge location, and the dispersion pattern is

more spread towards the northward side. The thermal dispersion of future scenarios has been analysed. Figures 16 and 17 shows the temperature after dispersion during different ambient conditions, namely strong wind (10 m/s) and calm wind conditions (4 m/s). Figure 18 shows the thermal plume dispersion when released further offshore from the existing discharge location. It is clearly evident that the temperature plume dispersion is quick, and the excess temperature is lower than that at the present discharge location.

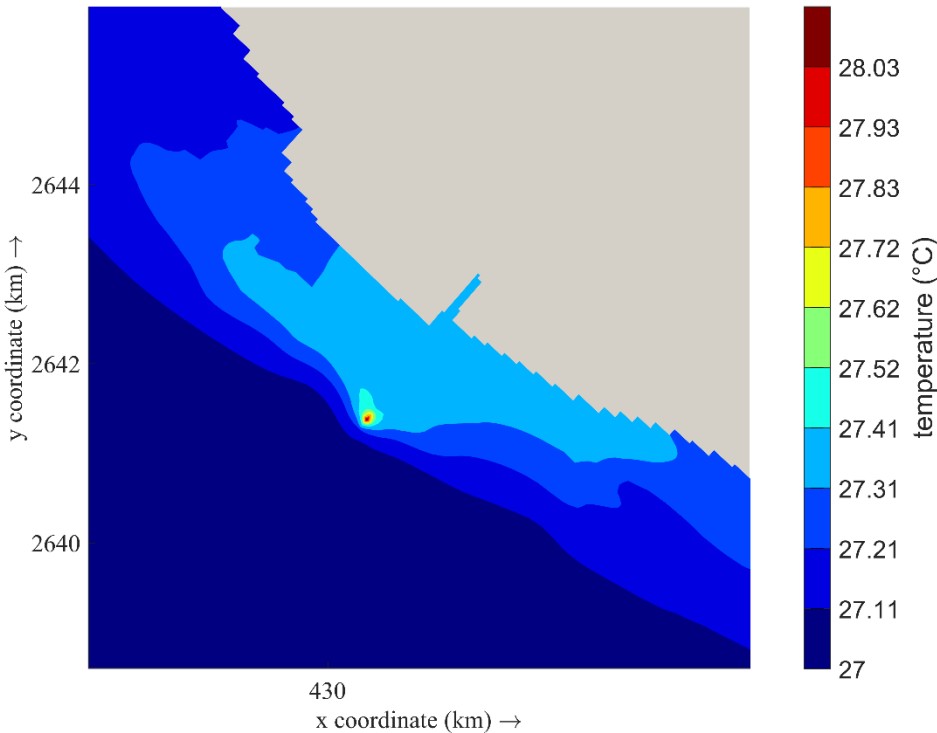

**Figure 18.** Thermal dispersion at the far discharge point.

It is worth noticing that the salinity plume dispersion is situated towards the northwest during the winter (Figure 15). An almost similar pattern is observed at the offshore point and the outfall location. At the proposed location, the discharge plume quickly dispersed, and the impact along the coast is very low, with the salinity plume near the coast always being less than 1 PSU. A similar condition is observed when the discharge location is further away from the coast, with the change in salinity being less than 0.5 PSU near the discharge location. In addition, it is well noted that the proposed offshore outfall and further-offshore locations are far enough to ensure a quick dispersion. The quick and impactless dispersion pattern of brine discharge ensures the least/negligible impact of salinity near the coast, enabling a safe environment near the coast.

The model simulations were performed during different seasons according to the scenarios considered. Within the proximity of the mixing zone, the plume is observed as vertically well mixed. When the discharge water is negatively buoyant, it will tend to sink towards the bottom and will spread laterally while being advected by the ambient current. At this stage, the plume thickness normally reduces. The effect of ambient velocity on the mixing rate is relatively strong.

Furthermore, regarding the thermal dispersion, the northward and southward plume pattern is clearly observed in all scenarios. This is clearly in line with the divergence and eddy circulation near the discharge point, as mentioned in the hydrodynamics section. Although the dispersion reaches the north and south of the coastal region, at calm wind condition Option 1, there is no significant increase in the temperature in the vicinity of the outfall. Here, the excess temperature due to the offshore outfall is less than 2 °C at the discharge location, while being negligible along the coast. For strong wind conditions,

there is a significant increase in the temperature of the nearshore waters close to the outfall. Still, the excess temperature here is less than 4 °C.

Although the movement of plumes near the coast is clearly visible, the plumes don't move further offshore because of the recirculation, and due to significantly better mixing with the ambient seawater, a higher dissipation is ensured. The offshore location of the existing discharge point also shows very little dispersion within the limits of EIA for thermal dispersion.

Furthermore, as demonstrated by the present study, the Yanbu discharge plume occurs predominantly during winds from westerly directions (normal and abnormal summer winds). During both normal and strong wind conditions, the intake position is not influenced. The average water temperature is expected to increase/exceed at least 4 °C within 500 m of the discharge point and 10 °C within 1000 m of the disposal location. During strong winds, the excess temperature is less than 4 °C for large areas along the coast, and during calm wind conditions, the thermal dispersion pattern suggests increases of less than 2 °C within 500 m and less than 10 °C at more than 1000 m.

## 8. Summary and Conclusions

A numerical modelling study has been conducted to simulate the hydrodynamics and processes of thermal and brine discharges at the desalination plant of Yanbu along the central-eastern coast of Saudi Arabia, as part of a strategic environmental impact assessment. The highlights of the study are as follows:

- Seasonal variability does play a significant role in the mixing and dispersion of wastewater plumes.
- Although the tidal flow is lower, the ambient current and wind causes the far-field discharge to spread along the north–south direction during the winter and summer.
- The modelling results also predicted the plume oscillate and modified its direction with each flood and ebb event; as a consequence of a change in the direction of the current, the plume moves in a north–south direction together with the presence of anticyclonic local eddies in the region.
- During the winter, the advection is mostly situated towards the south, and the spread of the plume moved southward to the discharge location.
- Furthermore, the model results show that the risk of recirculation at the present discharge location is minor. The region never experiences an unacceptable level of dilution. The presence of a strong flow near the discharge location and the tidal mixing could be the possible reason for this.

It is necessary to develop strategic planning and activities for the Kingdom's sustainable coastal development. Although ocean energy currently accounts for a very small proportion of the energy system, it has great potential for further development, considering that natural resource demand is exponentially increasing even though much natural potential exists for various sources such as wind, solar, and oil in the Kingdom. The conversion of ocean waves, currents, and tides into sustainable electrical power has been an increasing concern worldwide, and energy resource assessments therefore become essential. From this perspective, we are keen to support the country by providing reliable assessments of the marine renewable energies along the coast of Saudi Arabia.

Moreover, considering the Kingdom's Saudi Vision 2030 and taking this study as the baseline study of desalination plants along the coast, we will be working on the modelling of sediment transport dispersion and hydrodynamics all along the Saudi coastline to understand how the wave, current, and sediment dynamics act on the system.

As all the marine environmental ecosystems are preserved in the Saudi Vision 2030, it is a significant approach to predict the possibilities of environmental hazards with respect to hydrodynamic and biogeochemical modelling. The present study will be further extended to account for the effects of benthic organisms and their relationship with coastal wave energy using advanced modelling tools. Furthermore, the proposed model will test

the possibilities of further detecting coastal dispersion along the entire Red Sea coast by combining it with a biogeochemical model.

**Author Contributions:** N.S.A.: conceptualisation and original draft preparation. Y.N.K.: conceptualisation and original draft preparation. P.R.S.: investigation and modelling. R.K.A.-F.: review and editing. I.I.S.: review and editing. N.H.A.: investigation. A.J.T.: review and editing. M.A.S.: investigation, supervision, and review. All authors have read and agreed to the published version of the manuscript.

**Funding:** The work was funded by King Abdulaziz University and the Institutional Fund Projects under grant no. (IFPHI-181-130-2020).

**Institutional Review Board Statement:** Not Applicable.

**Informed Consent Statement:** Not Applicable.

**Data Availability Statement:** The data used to support the findings of this study are included within the article.

**Acknowledgments:** The authors gratefully acknowledge technical and financial support from the Ministry of Education and King Abdulaziz University, DSR, Jeddah, Saudi Arabia.

**Conflicts of Interest:** The current paper has no conflict of interest. The corresponding author approves the above statement as well.

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
