# Peer review of "Environmental Impacts of Thermal and Brine Dispersion Using Hydrodynamic Modelling for Yanbu Desalination Plant, on the Eastern Coast of the Red Sea"

_sustainability, doi:10.3390/su14084389_

Round 1

Reviewer 1 Report

The authors of the article "Environmental impacts of the thermal and brine dispersion using hydrodynamic modeling for Yanbu desalination plant, eastern coast of Red Sea" carried out a comprehensive study of the Red Sea coast near the Yanbu desalination plant. The simulation uses real data on the studied area, so the results are most likely reflects the processes that actually occur in the region. However, there are a few comments for the authors of the article:

1. What is the reason for the higher density regions on the grid in the center and along the coast in Figure 2?

2. It would be appropriate to include depth map of the studied area.

3. It would be nice to add a coastline to all figures for a better readability.

4. It would be nice to clarify the velocity of strong and calm wind in Item 7.

5. English could be improved.

Author Response

All the authors are grateful to this anonymous reviewer for his/her time to spent on this manuscript. All the suggestions are accepted and it helps to improve the quality of the manuscript. All the replies are attached and it is added in the track changed version of the manuscript.

Reviewer 2 Report

sustainability-1633740 - Environmental impacts of the thermal and brine dispersion using hydrodynamic modeling for Yanbu desalination plant, east
ern coast of Red Sea 

Current research paper shows Environmental impacts of the thermal and brine dispersion using hydrodynamic modeling for Yanbu desalination plant, eastern coast of Red Sea. Title of the article is very interesting but the content is not interesting. Grammatical and typo errors are there in the manuscript. Authors are also advised to reduce the plagiarism of paper. Kindly go through the following points and revise it accordingly:

  1. Abstract of the article needs to be improved
  2. Lumpy references are shown in the introduction section, i am advising authors to revise the introduction and remove lumpy references i.e.[19-29].
  3. Problem definition is not clear
  4. Objectives needs to be addressed properly
  5. section 2 must be elaborated properly
  6. ahead of the methodology dot is added by authors so kindly remove dot
  7. " Recent advances in the modeling has increased dramatically such that it can be effectively used as the tool for the many environmental issues" kindly add the citations.
  8. section 3.3 must be elaborated properly
  9. see the boundary conditions carefully
  10. see section 4.1 carefully
  11. see section 5.1 carefully
  12. snapshot is not proper word
  13. see figure 12 carefully
  14. discussion of figures 14 to 18 must be improved
  15. Conclusion needs to be refined and try to add points of conclusion in bullet points
  16. Add future work

Author Response

All the authors are thankful to this anonymous reviewer for his/her time to spent on this manuscript. All the suggestions are accepted and it helps to improve the quality of the manuscript. All the replies are attached and it is added in the track changed version of the manuscript.

Round 2

Reviewer 2 Report

All the comments have been fulfilled by the authors. But still, i am advising authors to see the grammar of the paper. Also advising authors to see the plagiarism part. 

Decision: Minor revision

Author Response

I would like to express my sincere thanks to the reviewer for his/her time on my manuscript. Of course, the suggestions were very useful and enhanced the quality of the manuscript. The reply to the new suggestions are incorporated in the revised manuscript and it is mentioned below,

Q- All the comments have been fulfilled by the authors. But still, i am advising authors to see the grammar of the paper. Also advising authors to see the plagiarism part. 

Reply: Thank you for your suggestions. We have fixed the English correction with the Native Speaker and done the plagiarism checking as well.
